# Bacterial Community Structure and Potential Microbial Coexistence Mechanism Associated with Three Halophytes Adapting to the Extremely Hypersaline Environment

**DOI:** 10.3390/microorganisms10061124

**Published:** 2022-05-30

**Authors:** Lei Gao, Yin Huang, Yonghong Liu, Osama Abdalla Abdelshafy Mohamed, Xiaorong Fan, Lei Wang, Li Li, Jinbiao Ma

**Affiliations:** 1State Key Laboratory of Desert and Oasis Ecology, Xinjiang Institute of Ecology and Geography, Chinese Academy of Sciences, Urumqi 830011, China; gaolei19@mails.ucas.ac.cn (L.G.); huangyin21@mails.ucas.ac.cn (Y.H.); liuyh@ms.xjb.ac.cn (Y.L.); osama@aru.edu.eg (O.A.A.M.); xiaorongfan@njeu.edu.cn (X.F.); egiwang@ms.xjb.ac.cn (L.W.); 2College of Resources and Environment, University of Chinese Academy of Sciences, Beijing 100049, China; 3School of Life Sciences, Key Laboratory of Microbial Diversity Research and Application of Hebei Province, Hebei University, Baoding 071002, China

**Keywords:** endophytic bacteria, rhizobacteria, diversity, community, halophyte, high-throughput sequencing

## Abstract

Halophytes play a crucial ecological role in drought and saline–alkali environments. However, there is limited knowledge about the structure of bacterial communities and the potential microbial coexistence mechanism associated with halophytes. This study investigated the diversity and community structure of endophytic and rhizospheric bacteria associated with three halophytes by applying high-throughput sequencing and geochemistry analyses on the studied soils. We collected 18 plant and 21 soil samples, and sequenced the V3 and V4 hypervariable regions of the 16S rRNA gene using next-generation sequencing (NGS). We also assessed geochemistry of the studied soils. The research suggested that rhizospheric bacterial richness and diversity associated with three halophytes were all significantly higher than for endophytic bacteria. The microbial community analysis indicated that *Actinobacteria*, *Firmicutes*, *Bacteroidetes* and *Proteobacteria* were the dominating bacterial phyla. Most unassigned operational taxonomic units (OTUs) implied that the microbes associated with halophytes contained abundant potential novel taxa, which are significant microbial resources. The high-abundance OTU phylogenetic tree supported the above views as well. Additionally, network analysis indicated that some conditional rare taxa (CRT) also might be keystone taxa during halophyte microbial community construction. The results of non-metric multidimensional scaling (NMDS) ordination analysis indicated significant dissimilarities in the microbial community among different sample groups. Sixty-two biomarkers were detected from seven different sample groups by linear discriminant analysis effect size (LEFSe) analysis. Microbial functions predicted based on phylogenetic investigation of communities by reconstruction of unobserved states (PICRUSt2) demonstrated that the abundances of nitrogen metabolism genes of endophytic bacteria were significantly higher than in rhizobacteria. Environmental factor analysis confirmed that different soil properties have different degrees of influence on the abundance and composition of the microbiota. To better adapt to the extreme hypersaline environment, halophytes could specifically recruit some plant beneficial bacterial taxa, such as nitrogen-fixing bacteria and extremely halophilic or halotolerant bacteria, to help them robustly grow and proliferate. All our preliminary results highlight microbial diversity and community related to halophytes grown on saline–alkali land of arid areas. Simultaneously, this work also advanced our further understanding of the halophyte microbiome associated with plants, and their role in plant adaptation to the extremely hypersaline environment.

## 1. Introduction

The plant microbiome, which colonizes all accessible plant tissues and ecological niches, is a varied taxonomically structured community of microorganisms found in healthy plants [1]. The advances in methodology and sequencing technology in the past 20 years have promoted the research of the plant microbiome, revealing the importance of various ecological and evolutionary forces forming plant microbiota. These findings proved the two-way interaction between plant and microbiome and clarified a complex chemical relationship between plant, microbiome and environment [2]. Environmental stresses have adverse effects on plant growth and productivity. The plant microbiome plays a vital role in plant adaptation and resistance to abiotic stresses. Plant beneficial microorganisms have the potential to produce phytohormones (indoleacetic acid and gibberellic acid), solubilize (phosphorus, potassium and zinc), bind nutrients, elicit plant defense reactions against pathogens and support plant growth under harsh environments [3]. Metaomics techniques, such as amplicon sequencing, metagenomics, metatranscriptomics and metaproteomics, describing the diversity of archaea and bacteria associated with plants grown under extreme conditions and resolving the different mechanisms of microorganisms promoting plant growth under abiotic stress, have been widely used to study complex processes involved in microbe-mediated stress alleviation in different plants growing in extreme environments [4].

In extreme environments, plants are affected by high or low temperature, extreme drought, soil pollutants, hypersaline–alkali, etc. [5,6,7]. As the largest arid area in northwest China, Xinjiang has numerous adverse ecological environmental effects such as drought, high temperature and salinity, which adversely affect plant growth and development [8]. Due to water shortage combined with global climate change and human activity, soil desertification and salinization have always been the significant environmental challenges in this arid land [9]. Nowadays, soil salinization is a growing global ecological issue [10]. Hypersalinity depresses the growth of plants, decreases species diversity and changes the community of the plant microbiome [11]. Nevertheless, a few halophytes have adapted to such an extremely hypersaline environment due to long-scale evolution. Halophytes are a group of salt-tolerant plants that survive and generate high biomass in high-salinity soil environments, such as saline semi-deserts, saline–alkali soil, swamps and seashores [12,13]. Halophytes are important in agricultural and ecological research, such as vegetation restoration of saline soil, biodiversity maintenance, phytoremediation and increasing crop productivity [14,15]. Increasing evidence supports the theory that endophytic and rhizospheric bacteria play a vital role in plant development as studies on plant–microbe co-evolution and interactions have advanced [16,17]. Halophytes harbor distinct microbial communities inside their various organs and ecological niches, which perform critical effects on plant growth, pathogen resistance and stress adaptation [18]. Under some adversity stresses, plants could even actively seek assistance from the microbiome, such as reshaping the beneficial microbiota to enhance adaptability [19]. Conversely, host plants can provide nutrients and niches for endophytic and rhizospheric bacteria [20,21,22]. The structure and diversity of the plant microbiome greatly vary in different host plant species, ecological niches, growth statuses and living conditions [23,24]. Therefore, the exploration of response to stresses as well as composition and diversity differences of the halophyte microbiome are of great importance to understand the interaction between the microbiome and host plants. While the research about endophytic and rhizospheric bacteria has gradually increased in recent years, there are still many extraordinary halophyte microbial resources stagnating in the blank period of research against the background of increasing global soil salinization issues. Therefore, it is necessary to conduct in-depth and comprehensive analysis on some high-performing halophytes’ adaptability from multiple angles with microbiota.

In this paper, we report the diversity and community structure of endophytic and rhizospheric bacteria associated with three dominant halophytes collected from the salt-affected soil in Wujiaqu, Xinjiang. The objectives of our study were as follows: (1) to wholly learn the community composition of endophytic and rhizospheric bacteria associated with three halophytes; (2) to compare the difference in microbial diversity and community composition among different halophytic ecological niches; (3) to reveal the effect of environmental factors on the microbial community; and (4) to resolve the adaptability of halophytes to extremely hypersaline environments from microbiome insights. The findings of this study will provide a scientific insight into the diversity of endophytic and rhizospheric bacteria associated with three halophytes and the improvement of the stability of the saline–alkali land ecosystem. These pieces of knowledge help us understand the impact of soil salinization on the ecological and environmental changes in the arid zone.

## 2. Materials and Methods

### 2.1. Study Location and Sampling Methods

Three dominant halophytes, which were identified as *Salicornia europaea* L. (abbr. P1), *Kalidium foliatum* (Pall.) Moq. (abbr. P2) and *Borsczowia aralocaspica* Bunge (abbr. P3), as well as their rhizospheric soil, were collected from the saline–alkali soil of Wujiaqu, Xinjiang, China in 2020 (Figure 1B). Three halophytes from one site at the same time were labeled as P1EB, P2EB and P3EB sample groups, respectively, and placed in aseptic bags which were put on ice immediately and transported back to our lab, and stored at 4 °C. Simultaneously, rhizospheric soil samples were labeled as P1RB, P2RB and P3RB sample groups, respectively, and placed in aseptic centrifuge tubes (50 mL) which were placed on ice straightaway, transported back to our lab and stored at −20 °C. In addition, we also collected open blank soil samples (labeled as OSB sample group) from the bare saline–alkali soil without plant growth to further explore the peculiarity of the plant microbiome. The information of all samples is shown in Appendix A. Soil properties were determined by the Xinjiang Institute of Ecology and Geography, Chinese Academy of Sciences (Appendix A).

### 2.2. Sterilization of Plant Materials

Sterilization procedures were completed for the collected halophyte samples before DNA extraction. Briefly, the whole plant was washed under running tap water to remove the soil attached to the root and dust on the plant surface. After initial washing, the plant samples were cut into 1–2 cm pieces by using sterile scissors. Subsequently, plant samples were washed by ultrasound for 15 min at 45 kHz to remove the tiny soil grains. The cleaned plant samples were sterilized with 75% ethanol for 1 min and with 5% NaClO for 8 min, then rinsed five times using sterile distilled water in a laminar airflow chamber [25]. To check the sterility of the surface of plants after surface sterilization of plant materials, we spread 100 μL of the last rinse of ddH_2_O on TSA and marine agar 2216 plates. After 7 days at 30 °C, the absence of colonies on the TSA and marine agar 2216 plates confirmed that plant epiphytic bacteria on the plant issues were successfully removed. With 48 h of air drying in a laminar airflow chamber, all of the sterilized plant samples were crushed by a sterile masher to extract DNA and stored at −20 °C for further experiments.

### 2.3. DNA Extraction, PCR Amplification, 16S rRNA Gene Clone Library Construction and Sequencing

The prepared soil and plant samples were sent to Biomarker Technologies, Beijing, China. Total plant and soil microbial DNA extraction, PCR amplification, 16S rRNA gene clone library construction and sequencing were all completed by Biomarker Technologies. In this study, there were 12 samples (6 EB samples and 6 RB samples) associated with P1, 12 samples (6 EB samples and 6 RB samples) associated with P2, 12 samples (6 EB samples and 6 RB samples) associated with P3 and 3 open blank soil samples (OSB), all of which (39 samples) were sequenced. The endophytic bacterial target-specific primers 335F (5′-CADACTCCTACGGGAGGC-3′) and 769R (5′-ATCCTGTTTGMTMCCCVCRC-3′) were used to amplify the V3 and V4 hypervariable regions of the bacterial 16S rRNA genes [26]. The soil bacterial target-specific primers 338F (5′-ACTCCTACGGGAGGCAGCA-3′) and 806R (5′-GGACTACHVGGGTWTCTAAT-3′) were used to amplify the V3 and V4 hypervariable regions of the bacterial 16S rRNA genes. After all of the samples were tested to be qualified, the Hiseq PE250 platform (Illumina, San Diego, CA, USA) was used for high-throughput paired-end sequencing of the purified amplicons.

### 2.4. Raw Sequence Data Processing

The raw data of 16S rRNA paired-end reads were cut with forward and reverse primers using QIIME2 plug-in Cutadapter [27,28]. The paired-end fastq files were merged and redundancy was removed using VSEARCH software [29]. Using unoise3 denoised, predicted biological sequences, filtered chimeras and effective tags were obtained [30]. A cluster of reads with 97 percent sequence similarity was identified as an operational taxonomic unit (OTU) to maximize the utilization of sequences. Each OTU was annotated with the SILVA high-quality ribosomal RNA database using USEARCH (ver. 10.0.240). After that, mitochondria and chloroplasts were deleted from our data using the QIIME2 (ver. 2021.11). 

### 2.5. Statistical Analysis

The dataset without singletons was rarefied to the minimum number of reads (45,874) recovered from our samples for comparative analysis of endophytic and rhizospheric bacterial richness and diversity indices (Observed_OTUs, Chao1, ACE, Shannon, Simpson, J, PD and Good_coverage) between sample groups, using R packages ‘MicrobiotaProcess’, ‘phyloseq’ and ‘microeco’. The Wilcoxon test was used to test for significant differences among alpha diversities. The UpSet diagrams were produced with the R package ‘Microbiota Process’. Venn and ternary plots were completed using the online website http://www.cloud.biomicroclass.com/CloudPlatform/home, accessed on 15 December 2021. The relationships between endophytic and rhizospheric bacterial community structures were appraised by NMDS in R package ‘phyloseq’. Furthermore, we used LEFSe to identify differentially abundant species among samples for biomarker discovery in different ecological niches. The 16S rRNA gene maximum likelihood (ML) phylogenetic tree was built with representative sequences of related high-abundance bacteria (top 100) using fasttree and displayed with the use of Interactive Tree of Life (iTOL). Additionally, we also used phylogenetic investigation of communities by reconstruction of unobserved states (PICRUSt2) to predict the function of endophytic and rhizospheric bacteria. R (version 4.1.3) and STAMP tools were used to accomplish the KEGG enrichment analysis and differential metabolic study. P values were corrected for multiple comparisons using the false discovery rate (FDR) with the Benjamini–Hochberg method. The redundancy analysis (RDA), correlation heatmap, linear regression, random forest (RF) and aggregated boosted tree (ABT) analysis of microbes with environmental factors were completed by Microeco bioinformatics cloud (https://www.bioincloud.tech/, accessed on 5 February 2022) and R (ver. 4.1.3). Variation partitioning analysis (VPA) was performed to determine the relative contributions of halophytic ecological niches, the measured soil properties and the interactions among these factors to the distribution of bacterial communities using the Lingbo MicroClass cloud platform (http://www.cloud.biomicroclass.com/CloudPlatform/home, accessed on 12 February 2022). Microbial co-occurrence network analysis was finished through molecular ecological network analysis (MENA). Other statistical analyses and visualizations were completed by Microsoft Excel 2019 and Chiplot (https://www.chiplot.online/, accessed on 20 February 2022).

## 3. Results

### 3.1. Diversity Analysis of Bacteria Associated with Three Halophytes

After read-quality filtering, denoising and clustering, 2,303,642 high-quality bacterial 16S rRNA gene sequences were successfully obtained from 39 soil and plant samples for the endophytic and rhizospheric bacterial community, and they were classified into 6021 OTUs (Appendix A). Alpha rarefaction curves (Appendix A), combined with the estimated Good_coverage values (Appendix A), suggested that the sequencing depths were sufficient to obtain a large majority of the bacterial diversity in the samples. The observed OTUs, ACE, Shannon and Simpson of the bacterial communities showed significant differences between endophytic and rhizospheric bacteria. It is observed that the alpha diversity indices of rhizospheric bacteria samples (P1RB, P2RB and P3RB) are universally higher than endophytic bacteria samples (P1EB, P2EB and P3EB). Additionally, compared with OSB, the richness of rhizospheric bacteria was significantly higher while the diversity of endophytic bacteria was generally lower (Figure 1A). A total of 6021 OTUs were detected across all libraries with 432 OTUs common to all samples, while the numbers of OTUs exclusive to the OSB, P1EB, P1RB, P2EB, P2RB, P3EB and P3RB groups were 117, 130, 33, 66, 43, 70 and 29, respectively (Figure 1C). There were 11 exclusive OTUs of the OSB, there were 18 exclusive OTUs of the halophyte P1, there were 12 exclusive OTUs of the halophyte P2, there were 10 exclusive OTUs of the halophyte P3 and there were 188 shared OTUs between OSB and three different halophytes (Figure 1D). Therefore, the differences in halophytic microbial diversity were closely related to halophyte species and ecological niches.

### 3.2. Microbial Community Analysis Associated with Three Halophytes

High-throughput sequencing revealed the composition of endophytic and rhizospheric bacterial communities in all sample groups. Forty-four phyla were identified for endophytic and rhizospheric bacteria in total. The relative abundances of the top 18 phyla of the endophytic and rhizospheric bacteria are displayed in Figure 2A. These bacterial OTUs were mainly affiliated with *Proteobacteria*, *Firmicutes*, *Bacteroides*, *Actinobacteria* and *Acidobacteria*. *Proteobacteria* accounted for a relatively higher proportion of endophytic bacteria than rhizospheric bacteria (Appendix A). However, the relative abundances of *Bacteroides*, *Actinobacteria* and *Acidobacteria* of rhizospheric bacteria samples were higher than that of endophytic bacteria. In addition, there are some rare bacterial groups in our samples, such as *Verrucomicrobia* and *Gemmatimonadetes*. The relative abundance of the top ten genera of the endophytic and rhizospheric bacteria is shown in Appendix A, in which a large proportion of OTUs are not annotated to the genus level. This suggested that the halophyte microbiome has a significant number of potential novel microbial resources requiring further efforts to excavate. The relative abundances of genera *Actinomycetales* and *Sphingobacteriales* were higher in rhizospheric bacteria samples (P1RB, P2RB and P3RB) than in endophytic bacteria samples (P1EB, P2EB and P3EB). Oppositely, the relative abundances of the genera *Staphylococcus*, *Kushneria* and *Enterobacter* were higher in endophytic bacteria than rhizospheric bacteria samples.

In Figure 2B, the representative sequences of the top 100 abundant endophytic and rhizospheric bacteria in this study were selected to construct the phylogenetic tree. Based on the phylogenetic relationship, it can be found that the microbiota of halophytes has the characteristic of high diversity. The top 100 abundant bacteria belonged to 9 phyla, 17 classes and 41 genera. Among these, 54 OTUs belonged to *Proteobacteria*, 17 to *Bacteroidetes*, 16 to *Firmicutes*, 7 to *Actinobacteria*, 2 to *Chloroflexi*, 1 to *Verrucomicrobia*, 1 to *Spirochaetes*, 1 to *Nitrospinae* and 1 to *Acidobacteria*. The phylogenetic tree also showed that the endophytic and rhizospheric bacteria of the three halophytes contained some unassigned potentially novel taxa. The phylogenetic tree also shows the OTU abundance of the top 100 taxa among different sample groups. In the OSB, we observed a high abundance of the genus *Halomonas*. In P1EB and P1RB, the genus *Enterobacter* was enriched. In P2EB, there were a relatively high proportion of unassigned taxa, indicating that many of the potential unknown taxa abound in this sample group. In P2RB, we found an increased enrichment of the genera *Rickettsia* and *Thalassospira*. Furthermore, the genus *Staphylococcus* was enriched in P3EB. Finally, in P3RB, we observed an apparent enrichment of the genus *Pantoe*.

The beta-diversity analysis based on NMDS of unweighted UniFrac distance was performed to compare the microbial community structure difference. The NMDS ordination analyses indicated that the bacterial community compositions differed significantly among the studied soil and plant samples (stress: 0.0785) (Figure 2C). Those samples belonging to the different sample types exhibited a modestly strong separation, while those samples belonging to the same sample type failed to be significantly separate. The endophytic bacteria sample groups (P1EB, P2EB and P3EB) and rhizospheric bacteria sample groups (P1RB, P2RB and P3RB) of the three halophytes, as well as OSB, can be well separated. In addition, the endophytic bacterial community structure between P1EB and P3EB was also obviously different. However, the rhizospheric bacterial community structure associated with three halophytes showed no apparent difference.

To illustrate the intergroup differences of bacteria among different ecological niches and species of halophytes, ternary plots were employed to display the taxonomic information of different OTUs at the phylum level. The differences among the three halophytes showed that most of the high-abundance taxa have little difference among the three halophytes. In contrast, some low-abundance taxa are enriched in a specific plant microbiome (Figure 2D). For example, phyla *Chlorobi*, *Fibrobacteres*, and *Ignavibacteriae* were only detected in the halophyte P1 microbiome (Appendix A). Interestingly, the phylum *Chlorobi*, as a group of obligately anaerobic photoautotrophic bacteria, could reduce nitrogen to ammonia as previously described, which indicated that this taxon had the potential to promote the growth of the halophyte P1, *S. europaea* L. [31,32]. The differences between different ecological niches showed that most of the high-abundance taxa have little difference among different environmental niches, while some low-abundance taxa are enriched in the rhizospheric microbiome, such as *Euryarchaeota* and *Fibrobacteres* (Figure 2E and Appendix A). The differences between different P1, P2 and P3 plant niches as well as the OSB group showed that most of the high-abundance taxa have little difference among different ecological niches. At the same time, some low-abundance taxa are also enriched in rhizosphere microbiome, such as *Euryarchaeota* and *Fibrobacteres* (Figure 2F–H as well as Appendix A). It is noteworthy that the phylum *Euryarchaeota* includes a taxon of the class *Halobacteria*, which survives extreme concentrations of salt [33]. This means that this taxon may play an important role in halophytes’ salt and alkali tolerance.

In a word, the distinct differences in microbial communities associated with halophytes were detected between different halophyte species and ecological niches. Moreover, the dominant microbial taxa with high abundance have little difference between different ecological niches and halophytes, while some rare microbial taxa with low abundance have significant differences. These low-abundance taxa may play a decisive role in different halophyte ecological niches.

### 3.3. Halophytic Microbiome Co-Occurrence Network Analysis

The relationships between microbial taxa also shape the structure of microbial communities [34] and, thus, we constructed co-occurrence network patterns using the whole OTU, conditionally rare taxa (CRT) and conditionally rare or abundant taxa (CRAT) datasets, respectively [35], based on MENA. The co-occurrence network of the whole OTU dataset consisted of 224 nodes (OTUs), 358 edges (average degree or node connectivity 3.196; average path distance 5.293) and 18 modules. Moreover, the node with the max degree, betweenness, stress centrality and eigenvector centrality is OTU_870, which is affiliated with the phylum Proteobacteria and included in conditional rare taxa (Figure 3A and Appendix A). This shows that OTU_870 played a crucial role in constructing the halophyte microbial community. At the same time, it also further verified the previous conclusion that some rare or conditionally rare taxa could also be keystone taxa in this process in the construction of the microbial community, except rich, CAT and CRAT microbial taxa with high abundance [36]. The co-occurrence network of the CRT dataset consisted of 231 nodes (OTUs), 389 edges (average degree or node connectivity 3.368; average path distance 4.742) and 31 modules. Moreover, the node with the max degree and stress centrality is OTU_322, which is affiliated with the phylum Actinobacteria (Figure 3B and Appendix A). The co-occurrence network of the CRAT dataset consisted of 69 nodes (OTUs), 1112 edges (average degree or node connectivity 32.232; average path distance 1.526) and 2 modules. This indicated that the interaction between the CRAT taxa is closer (Figure 3C and Appendix A).

### 3.4. Biomarker Analysis of Different Ecological Niches

Significant abundance differences were detected in the bacterial community compositions among seven sample groups (Figure 4). Sixty-two taxa with significantly different abundances were described among the various sample groups, according to the LEFSe pipeline (LDA > 4, *p* < 0.05). In detail, twenty-two taxa were enriched in the OSB sample group, such as *Salinisphaera*, *Bacteroidetes*, *Flavobacteriales*, etc. Eight taxa were enriched in the P1EB sample group, such as *Gammaproteobacteria*, *Kushneria*, *Spirochaetaceae*, etc. Fourteen taxa were enriched in the P1RB sample group, such as *Lactobacillus*, *Actinobacteria*, *Lactobacillaceae*, etc. Thirteen taxa were enriched in the P2EB sample group, such as *Streptococcaceae*, *Bacillales*, *Prevotella*, etc. Twenty-six taxa were enriched in the P2RB sample group, such as *Cytophagia*, *Verrucomicrobia*, *Acidobacteria*, etc. Twelve taxa were enriched in the P3EB sample group, such as *Rickettsiaceae*, *Microbulbifer*, *Rhodospirillales*, etc. Ten taxa were enriched in the P3RB sample group, such as *Chromatiales*, *Sphingobacteriales*, *Ectothiorhodospiraceae*, etc. These significantly different biomarkers may be important sources of differences in microbial community structure among different halophyte ecological niches.

### 3.5. Predicted KEGG Pathways of Endophytic and Rhizospheric Bacteria Based on PICRUSt2

All bacterial functions were predicted using the PICRUSt2 algorithm. A total of 47 relevant KEGG categories were expected, including Cellular processes, Environmental information processing, Genetic information processing, Human diseases, Metabolism and Organismal systems. Amino acid metabolism, Carbohydrate metabolism, Metabolism of cofactors and vitamins and Metabolism of other amino acids were the four most enriched KEGG pathways (Figure 5). The KEGG pathways of the halophyte microbiome related to membrane transport also have a high abundance to adapt to hypersaline. To further determine the function abundance difference in nitrogen metabolism, we analyzed and compared the abundance of the KEGG pathway related to nitrogen metabolism between sample groups. We found the abundance of nitrogen metabolism is universally higher in the endophytic bacterial sample groups than in rhizospheric bacterial sample groups (Appendix A). This finding probably indicates that the higher nitrogen metabolism (N-fixing) taxa are distributed in halophyte tissues, especially in roots. The difference analysis of functional abundance between different sample groups based on the PICRUSt2 function prediction results by the STAMP software showed that there are a total of 41 functional genes showing differential abundance between P1EB and P1RB, a total of 9 functional genes showing differential abundance between P2EB and P2RB and a total of 63 functional genes showing differential abundance between P2EB and P2RB. All detailed functional genes showing differential abundance are listed in Appendix A.

### 3.6. Influences of Different Soil Properties on Microbial Community

According to the RDA results, it could be seen that soil properties have different effects on microbial communities of varying halophyte ecological niches (Figure 6A). There were positive correlations between the microbial communities of the OSB sample group and some environmental factors such as Ca^2+^, Cl^−^, Mg^2+^, K^+^, Na^+^, SO_4_^2−^, HCO_3_^−^, conductivity and total salinity. The heatmap of the correlation between the top 30 genera and soil environmental factors (Appendix A) revealed that the genus *Lactobacillus* has a significantly positive correlation with available phosphorous. In contrast, the genus *Lactobacillus* has an extremely negative correlation with HCO_3_^−^ and total nitrogen. *Lactobacill**us*, with acid resistance to some extent, is found in a wide variety of environments, including soil (most commonly associated with the rhizosphere), plants (particularly decaying plant material) and animals [37]. Due to HCO_3_^−^ and ammonium nitrogen being alkaline, a negative correlation between *Lactobacillus* and the above two soil properties was considered reasonable. In addition, the genus *Sphingomonas* also displayed a significantly negative correlation with HCO_3_^−^.

Furthermore, *Gracilimonas*, a potential organic pollutant-degrading bacteria, had a significantly negative correlation with Ca^2+^. Again, the bacterial community dissimilarity showed a significant negative correlation (*p* < 0.05) with the content of Cl^−^, Na^+^, total potassium, total phosphorus and total salinity (Appendix A). The measured variables (such as different halophytes and ecological niches as well as other soil properties) could explain 26.35% of the observed bacterial community variations (Figure 6B). Among them, soil properties could explain 18.63% of the relevant microbial community structure variation alone. The remaining 73.65% of unexplained bacterial community variations may be affected by factors we failed to collect, such as human and livestock activities, etc. The effects of different soil properties on the microbial community composition associated with three halophytes were studied using ABT. Figure 6C shows the relative importance of these indicators. Na^+^ plays a dominant role (15.89%) in the microbial community structure stabilization associated with three halophytes among these environmental factors. To disentangle the potential main drivers of soil properties in saline–alkali soil ecosystems, we identified the main microbial predictors for the soil multinutrient cycling index by random forest (RF) analysis. Bacterial beta-diversity (MDS1) was the most essential variable for predicting the soil property ammonium nitrogen and total phosphorus cycling index. Bacterial alpha-diversity (PD) was the most crucial variable for predicting the soil property Ca^2+^, SO_4_^2−^ and total nitrogen cycling index.

Bacterial alpha-diversities (J and Shannon) were the most critical variable for predicting the soil’s total salinity cycling index. Detailed prediction findings of other soil properties are listed in Appendix A. We also evaluated the biological contributions of all microbial phyla to soil properties via an RF analysis. Evidently, not all microbial phyla contributed similarly to the various edaphic variables. For example, *Chloroflexi*, *Deinococcus*-*Thermus*, *Latescibacteria*, *Microgenomates* and *Verrucomicrobia* were the most important variables for predicting soil property Na^+^, as one of the main ions in soil properties of saline–alkali land, indicating their importance in soil property cycling during re-vegetation to some extent (Appendix A). Generally, soil properties have a strong correlation with the microbiome diversity and communities associated with halophytes. The changes in halophyte microbiome diversity and community structure will affect the fluctuation of environmental factors. The changes in corresponding ecological factors will also affect the halophyte microbiome diversity and community structure.

### 3.7. Microbial Insights about Halophytes Adapting to the Extreme Hypersaline Environment

*Bacteroidetes* could colonize a variety of habitats on Earth, such as rhizospheric soil in various locations, including cultivated fields, greenhouse soils, unexploited areas, etc. The halophilic genus *Salinibacter* from the phylum *Bacteroidetes* always lives in salt-saturated brines and hypersaline soils. *Salinibacter* shares many properties (especially strong salt tolerance) with halophilic archaea such as *Halobacterium* and *Haloquadratum* that inhabit the same environments [38]. *Salinibacter* was also detected in our collected halophyte rhizospheric soil as well (Appendix A). The results of correlation analysis between environmental factors and relative abundance of the dominant phylum *Bacteroidetes* indicated that the correlations between Cl^−^, Mg^2+^, conductivity (also one of the indicators of soil salt content), as well as total salinity, and the relative abundance of the dominant phylum *Bacteroidetes* were significantly positive (Figure 7). They indicated that this taxon might represent a group of plant probiotics that can help plants cope with salt stress.

As can be seen from Figure 8A, the total salinity and Na^+^ contents of OSB soil are higher than rhizospheric soil, related to halophytes. This might be because halophilic microbial taxa in the microbial community associated with halophytes lower the salt content in the soil around the plants, allowing the plants to adapt to the highly hypersaline soil environment. Halophilic *Actinomycetes* often live in a hypersaline environment such as saline lakes and saline–alkali land. They frequently enrich some small molecular inorganic compounds in cells to adapt to the hypersaline environment and maintain osmotic pressure [39]. Our findings revealed that the abundances of several halophilic *Actinomycetes* were higher in microbial communities associated with three halophytes than OSB, such as *Nesterenkonia*, *Norcardiopsis*, *Pseudonocardiaceae* and *Streptomonospora*, indicating that these taxa could also play a crucial role in promoting halophyte growth and development under saline–alkali stress (Figure 8B). As can be seen from Figure 8C, the halophytic rhizosphere soil total nitrogen levels were higher than OSB. Our hypothesis is that the enriched nitrogen-fixing bacteria in the halophyte microbiome can increase the rhizospheric soil nitrogen content and promote plant growth. *Rhizobium* and *Klebsiella* can be found in a variety of plants, and are able to transform atmospheric nitrogen into a form that can be used by plants, and thus are called associative nitrogen fixers or diazotrophs [40,41]. Our findings revealed that the genera *Rhizobium* and *Klebsiella* were found only in microbial communities associated with three halophytes, and not in OSB, indicating that these taxa may play a crucial role in promoting halophyte growth and development under saline–alkali stress (Figure 8D).

In addition, we also compared and analyzed plants’ beneficial bacteria associated with different halophytes and ecological niches at the OTU level. The genus *Actinoplanes* has the potential to produce IAA, IPYA and GA3, promoting the growth of host plants as reported previously [42]. The genus *Rhizobium* has N2 fixation, phosphate solubilization, IAA production, siderophore production and ACC-deaminase activity, which are common beneficial traits for plants [41,43]. *Klebsiella* can be found in a variety of plant hosts, and can fix atmospheric nitrogen into a form that can be used by plants, and thus called associative nitrogen fixers or diazotrophs [40]. We also detected the above three taxa at the OTU level in our study. The abundances of these taxa were higher in the plant microbiome than in the OSB sample group (Figure 9A). Meanwhile, these plants’ beneficial bacteria were more enriched within the plant tissue (EB) than rhizospheric soil (RB) (Figure 9B–D). This result further supported the prediction of PICRUSt2. Our findings also showed that plant endophytic beneficial bacteria are more important for the adaptability of halophytes against adversity. To sum up, halophytes and specific ecological niches will specifically recruit some plants’ beneficial bacterial taxa such as those carrying out nitrogen fixation and with salt tolerance to help halophytes better adapt to the highly adverse high-saline environment. We also looked into the diversity of cultivable endophytic bacteria associated with these three halophytes in another study. Moreover, we have isolated some of the above-mentioned potential functional strains. Thus, further functional verification investigations will be planned with pot experiments in the future.

## 4. Discussion

The microbial diversity and community structure associated with three halophytes were investigated by the Illumina high-throughput sequencing of the V3 and V4 hypervariable regions of bacterial 16S rRNA genes. Based on diversity analysis, the diversity and richness of rhizospheric bacteria associated with three halophytes were higher than endophytic bacteria (*p* < 0.05, Figure 1A). Previous studies have confirmed that rhizospheric bacterial community compositions exhibit higher diversity and complexity than endophytic bacteria [44,45]. Similarly, the PLFA analysis of endophytic and rhizospheric bacteria associated with the roots of the halophyte *Aster tripolium* L. showed that the total bacterial biomass was the highest in the rhizosphere, followed by soil microorganisms, and the biomass of endophytic bacteria was the lowest [46]. Therefore, the distribution of microorganisms in different niches related to halophytes is very different, and the diversity of rhizospheric bacteria is generally higher than endophytic bacteria. The study on rhizosphere and endosphere bacterial diversity associated with two halophytes, *Glaux maritima* and *Salicornia europaea*, using next-generation sequencing (NGS) indicated that there are apparent differences in bacterial community composition and diversity between different plants and ecological niches [47]. One explanation for the higher abundance of root microbes may be that the roots secrete many secondary metabolites, which could be used as a supply of nutrients to cause microbial aggregation in the rhizosphere. In addition, some studies have found a close relationship between rhizosphere microorganisms and root exudate metabolites. A series of secondary metabolites secreted by roots can induce and change the diversity and community structure of root-associated microorganisms [48]. If the microbial composition in the rhizosphere changes, specific root exudates also will be induced [49].

The dominant phyla included *Proteobacteria*, *Firmicutes*, *Bacteroidetes* and *Actinobacteria*, according to the findings of the microbial community composition study. Additionally, the relative abundance of the dominant phyla varied consistently between the rhizosphere and endophytic tissues. The relative abundance of Proteobacteria was higher in endophytic bacteria, while that of *Bacteroidetes* and *Actinobacteria* was higher in rhizosphere bacteria (Figure 2A and Appendix A). An earlier analysis of the composition of the rhizosphere microbiomes of the halophytes with that of the non-halophytes showed that *Actinobacteria* was predominant in saline soil samples and *Firmicutes*, *Acidobacteria*, *Bacteroidetes* and *Thaumarchaeota* were all predominant among saline and non-saline soils [50]. Different from our results, they found that *Proteobacteria* was the dominant phylum in non-saline soil samples [50].

Furthermore, *Chloroflexi*, as a class of photosynthetic bacteria, was also detected in our samples (Figure 2A). Through high-throughput sequencing, metabolome and network analysis, Xian et al. found that *Tepidimonas* could secrete a large variety of complex metabolites containing a large variety of bacterial growth-promoting materials, which have the potential to promote the culturing of many previously uncultivated bacteria, such as *Chloroflexus*, eventually realizing the directional isolation and cultivation of uncultured *Chloroflexus* in a hot spring habitat. This provided new insights for us to explore microbial dark matter and resources in extreme environments. At the genus level, many OTUs are still unassigned. The bacterial abundance and composition are quite different within the different sample groups, which indicated that a great many novel taxa abound in our samples. The bacterial community has particular host specificity, in agreement with previous studies about other plants with culture-independent methods (Appendix A) [51,52]. The analysis of endophytic and rhizospheric bacterial community structure of the coastal halophyte *Messerschmidia sibirica* demonstrated that coastal halophytes display complex microbial communities and high diversity [45]. Additionally, microbial community composition analysis also showed that some rare microbial taxa with low abundance have great differences between different ecological niches and halophytes (Figure 2E–H). Furthermore, the co-occurrence network of the whole OTU dataset showed that some conditional rare taxa play an essential role during the microbial community construction (Figure 3A).

The phylogenetic analysis also found there was a diverse repertoire of endophytic and rhizospheric bacteria associated with three halophytes (Figure 2B). In addition to further understanding the endophytic and rhizospheric bacterial diversity and community structure, an increasing number of comprehensive studies into endophytic and rhizospheric bacterial diversity are very important and necessary for elucidating the roles of these bacteria and exploring these bioresources. Recently, many studies have potentially found plant-associated microbes to be a dominant factor in plant health and development. Bibi et al. (2018) isolated and identified 554 endophytic and rhizosphere bacteria associated with three halophytes grown in Saudi Arabia as well as screened out 57 fungal pathogen-resistant strains [53]. Based on both culture-dependent and culture-independent methods, Sheng et al. provided novel insights into the bacterial community, diversity and function related to the coastal halophyte *Limonium sinense*. They found that the genus *Glutamicibacter* has multiple potentials to promote plant growth and resist salt stress [54]. In the study of bacterial diversity related to the rhizosphere of halophytes in Pakistan, researchers found seven Bacillus-like bacterial genera, *Bacillus*, *Halobacillus*, *Virgibacillus*, *Brevibacillus*, *Paenibacillus*, *Tumebacillus* and *Lysinibacillus*, detected by using high-throughput sequencing, whereas only four Bacillus-like bacterial genera, *Bacillus*, *Halobacillus*, *Oceanobacillus* and *Virgibacillus*, were isolated by pure culture [55]. The research of microbial community structure and ecological function associated with the superior halo-tolerant *Suaeda salsa* found that these bacterial genomes include abundant genes contributing to salt stress acclimatization, nutrient solubilization and competitive root colonization to enhance plant stress fitness [56]. The NMDS analysis showed that bacterial community structures varied enormously between rhizospheres and endophytic tissues, while bacterial community structures among the same sample groups are similar (Figure 2C). Many pieces of research can also elucidate this point and mostly keep in line with our findings [45,51]. In brief, the differences in microbiome diversity and communities associated with halophytes were closely related to halophyte species and ecological niches. Moreover, the dominant microbial taxa with high abundance have little difference between varying ecological niches and halophytes, while some rare microbial taxa with low abundance have significant differences. In future further research, one of our research works is exploring a culture-dependent method that will be used to isolate potential novel bacteria and functional bacteria with plant growth promotion (PGP), stress resistance, enzyme production, etc. from halophytes. Exploring the diversity of cultivable endophytic bacteria in halophytes also has practical significance.

The comparison of these two studies determines that this phenomenon could be explained via the different plant effects. The LEFSe analysis revealed that sixty-two taxa with significantly different abundances were found among seven sample groups (Figure 4). To date, many studies have been using the LEFSe to find biomarkers among different sample groups associated with halophytes with abundance differences [57,58]. The predicted KEGG pathways of endophytic and rhizospheric bacteria based on the PICRUSt2 included 44 related categories and significantly differed among different plant ecological niches (Figure 5, Appendix A). A further study is necessary for the metagenomics to verify the KEGG pathways of the endophytic and rhizospheric bacteria related to three halophytes.

The soil properties in the halophyte habitat are closely related to the diversity and community structure of rhizosphere bacteria and endophytic bacteria (Figure 6 and Appendix A). On the one side, halophytes’ microbiome may change the ion composition of soil around the root through microbial metabolism to further affect the adaptability against adversity such as hypersaline stress. On the other side, the root network also allows halophytes to recruit distinct bacteria from a larger soil microbial reservoir to construct microbiomes to benefit themselves [59]. Li et al. confirmed the critical role of salt-induced root-derived bacteria (RDB) in enhancing plant adaptability to salt stress by measuring the composition and variation in the rhizosphere and endophyte bacteria of salt-sensitive (SS) and salt-resistant (SR) plants under soil conditions with or without salinity [60]. Our experimental results could also support this conclusion. Nitrogen-fixing bacteria (*Klebsiella* and *Rhizobium*), plants’ beneficial bacteria (*Actinoplanes*, *Klebsiella* and *Rhizobium*) and some halophilic bacteria (*Nesterenkonia* and *Pseudonocardiaceae*) are less often detected in the open blank saline soil without plant growth, while they can be detected in high abundance in the ecological niches related to halophytes (Figure 8B,D). A class of halophilic bacteria, *Pseudonocardiaceae*, have a relatively high abundance (Figure 8D).

Additionally, some plants’ beneficial bacterial taxa showed similar distribution patterns among different ecological niches (Figure 9). This could tentatively imply that halophytes could even actively reshape the beneficial microbiota to enhance adaptability under some adversity stresses. We have preliminarily obtained several halophilic and halotolerant strains in our ongoing experiment of mining halophilic microbial resources, which may be used to strengthen host plants’ stress resistance in future investigations. In the future, the exploration of culture-dependent methods for isolating bacteria from halophytes also needs to be combined with the results of environmental factor analysis to make up the best media. The contribution of this study is that, after we deeply understand the construction mechanism of the halophyte microbiome, we can design a set of microbial agents in reverse using our existing strain resources to improve plant stress resistance for agricultural production and solve the problems of insufficient cultivated land and decline in land fertility.

## 5. Conclusions

Based on the high-throughput sequencing methods, this study clarified the endophytic and rhizospheric bacterial diversity and community structure of *S*. *europaea* L., *K*. *foliatum* (Pall.) Moq. and *B*. *aralocaspica* Bunge collected from Wujiaqu, Xinjiang. The diversity and richness of rhizospheric bacteria related to three halophytes are significantly higher than endophytic bacteria. The dominant bacterial phyla associated with three halophytes were *Proteobacteria*, *Firmicutes*, *Bacteroidetes* and *Actinobacteria*. The dominant bacterial genera associated with three halophytes were Gracilimonas and Halomonas. The dominant endophytic bacteria of the three halophytes have little difference, while some rare microbial groups with low abundance have great differences. Moreover, some conditional rare taxa may also be keystone taxa during the halophytic microbial community construction. We also demonstrated that the bacterial diversity and community structure varied across the different ecological niches associated with halophytes, such as plant endogenous tissues, rhizospheric soil and open blank soil. Phylogenetic analysis showed that the microbiome associated with three halophytes exhibited high diversity and some unassigned potential novel taxa. Additionally, PICRUSt2 findings indicated that the abundance of nitrogen metabolism is universally higher in the endophytic bacterial sample groups than in rhizospheric bacterial sample groups. The results of environmental factor analysis made clear that different environmental factors have different degrees of influence on the microbial community composition and richness associated with three halophytes. The halophytes may specifically recruit and reshape beneficial microbiota taxa around different plant ecological niches, which could help host plants adapt to extremely hypersaline environments. Plant endophytic beneficial bacteria are more important for the adaptability of halophytes against adversity. Our findings provide some insights into the complex microbial community structure related to three halophytes collected from saline–alkali soil. Further studies are necessary to verify the function of these microbes in plant–microbe interactions in saline–alkali land using metagenomics sequencing and culture-dependent methods. It is also essential to isolate, purify and screen the strains with special functions by the pure culture method. Their functions will need to be verified via pot experiments or field experiments.

## Figures and Tables

**Figure 1 microorganisms-10-01124-f001:**
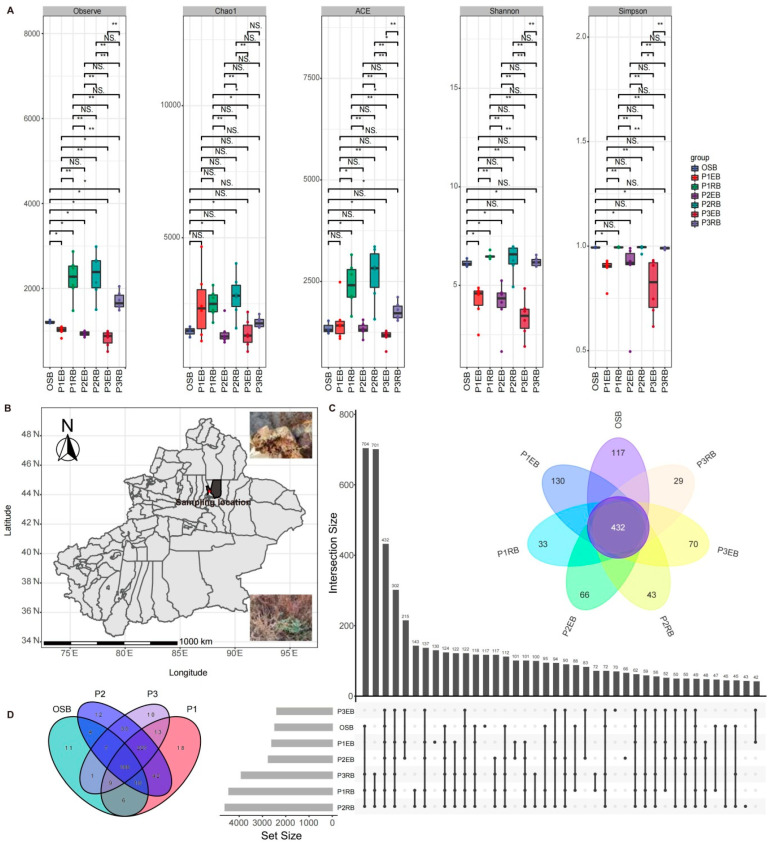
(**A**) The significant difference (Wilcoxon test, *p* < 0.05) in alpha diversity of endophytic bacteria and rhizospheric bacteria associated with three halophytes; (**B**) a map showing sampling location; (**C**) the UpSet view and flower plot showing the shared and special OTUs among different samples; (**D**) the Venn plot showing the shared and exclusive OTUs between OSB and three different halophytes. Note: NS. means no significant differences, * *p* < 0.05, ** *p* < 0.01.

**Figure 2 microorganisms-10-01124-f002:**
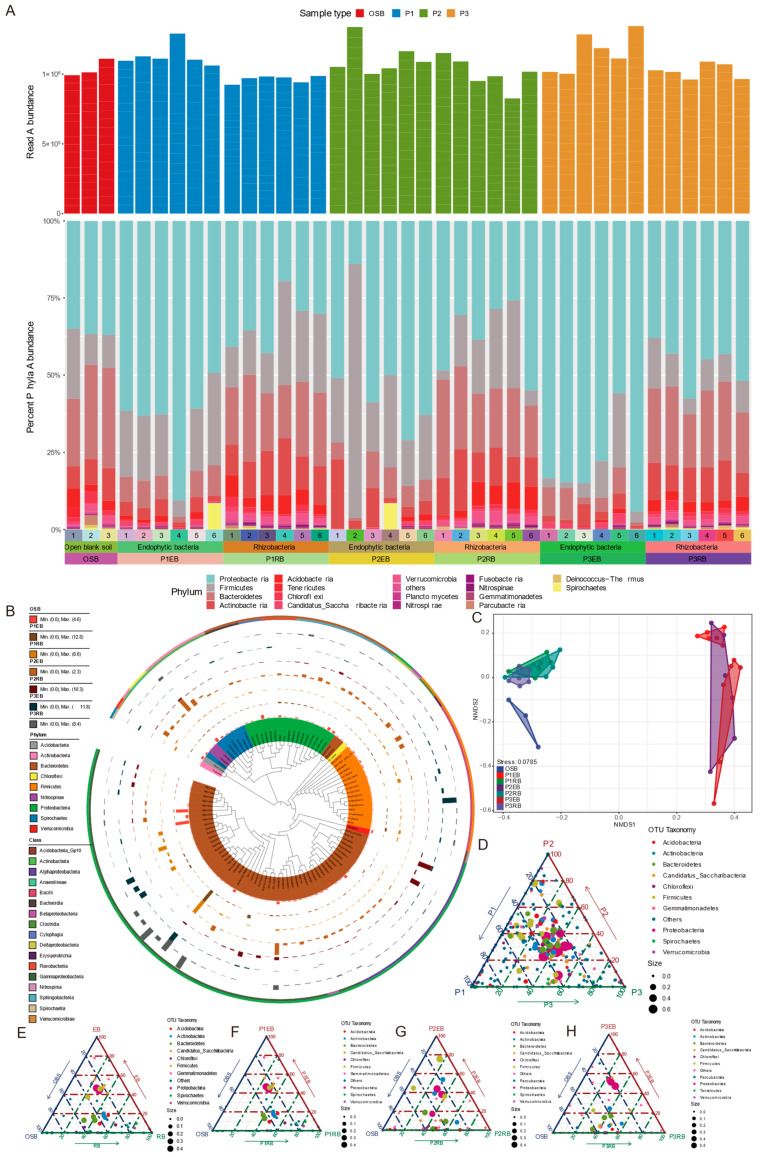
(**A**) Relative abundances of endophytic and rhizospheric bacteria at the phylum level (top eighteen) in different samples. (**B**) Taxonomic dendrogram showing top 100 abundant OTU members of the endophytic bacteria and rhizospheric bacteria. Color ranges identify phyla within the tree. Colored bars represent the abundance of each OTU in the different sample groups. The colored lines show the phyla and classes of the corresponding OTU. The taxonomic dendrogram was generated with one representative sequence of each OTU using QIIME2 and displayed with the use of Interactive Tree of Life (iTOL). (**C**) NMDS ordination based on unweighted UniFrac distance showing the difference in bacterial community composition among the studied soil and plant samples of Wujiaqu in this study. (**D**) The different OTUs between three different halophytes. (**E**) The different OTUs among different ecological niches. (**F**) The different OTUs between different P1 plant niches and the OSB. (**G**) The different OTUs between different P2 plant niches and the OSB. (**H**) The different OTUs between different P3 plant niches and the OSB. Notes: P1, *S*. *europaea* L., P2, *K*. *foliatum* (Pall.) Moq., P3, *B*. *aralocaspica* Bunge, EB, endophytic bacteria, RB, rhizobacteria, OSB, P1EB, P1RB, P2EB, P2RB, P3EB and P3RB mean the same as in the main text.

**Figure 3 microorganisms-10-01124-f003:**
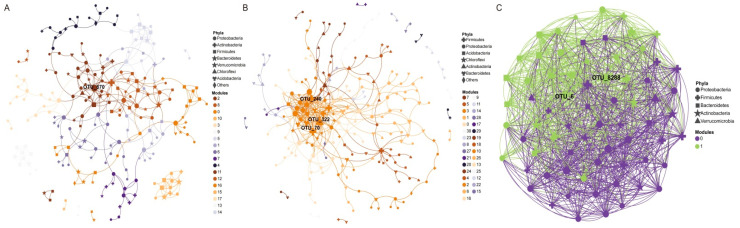
(**A**) Co-occurrence network of the whole dataset based on correlation analysis; (**B**) co-occurrence network of the CRT dataset based on correlation analysis; (**C**) co-occurrence network of the CRAT dataset based on correlation analysis. The size of each node is proportional to the number of connections (that is, degree). OTUs colored according to different modules. The shape of OTUs shows the taxonomy at the phylum level.

**Figure 4 microorganisms-10-01124-f004:**
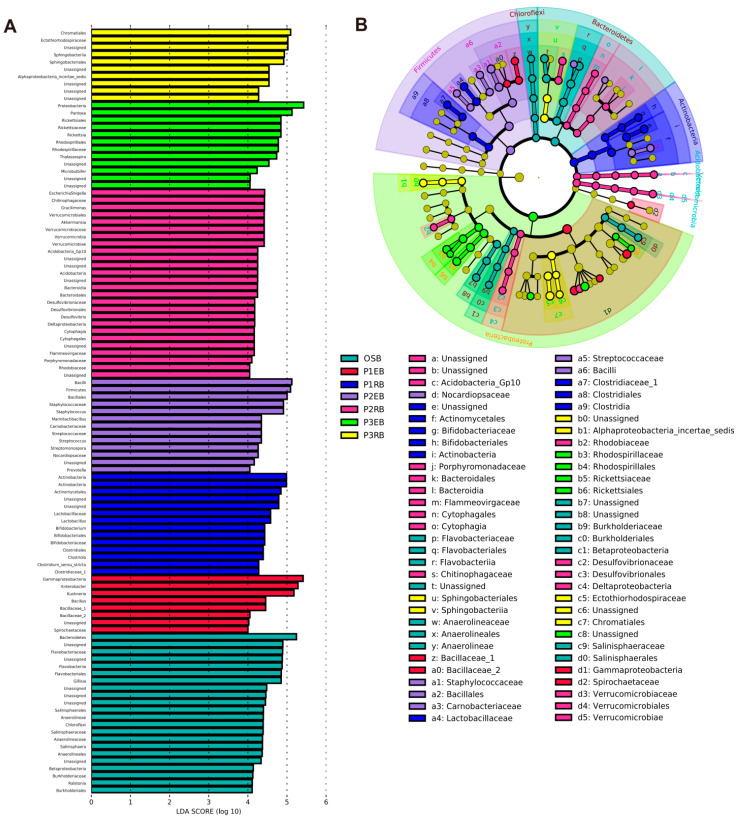
The bacterial taxa with differential abundance among 7 different sample groups, based on LEfSe software analysis. (**A**) Identified biomarkers ranked by the effect size in different samples. The habitat biomarkers were identified as being significantly abundant (LDA > 4, *p* < 0.05) when compared among samples. OSB refers to the open bare soil bacteria group, P1EB refers to the group of endophytic bacteria associated with P1, P1RB refers to the group of rhizospheric bacteria associated with P2, P2EB refers to the group of endophytic bacteria associated with P2, P2RB refers to the group of rhizospheric bacteria associated with P2, P3EB refers to the group of endophytic bacteria associated with P3, and P3RB refers to the group of rhizospheric bacteria associated with P3. (**B**) A cladogram representing the hierarchical taxonomic structure of the identified habitat biomarkers generated using LEfSe is shown. Each ring represents a taxonomic level, with phylum, class, order and family emanating from the center to the periphery. Each circle is a taxonomic unit found in the dataset, with circles or nodes shown in color where the taxon represented a significantly more abundant group.

**Figure 5 microorganisms-10-01124-f005:**
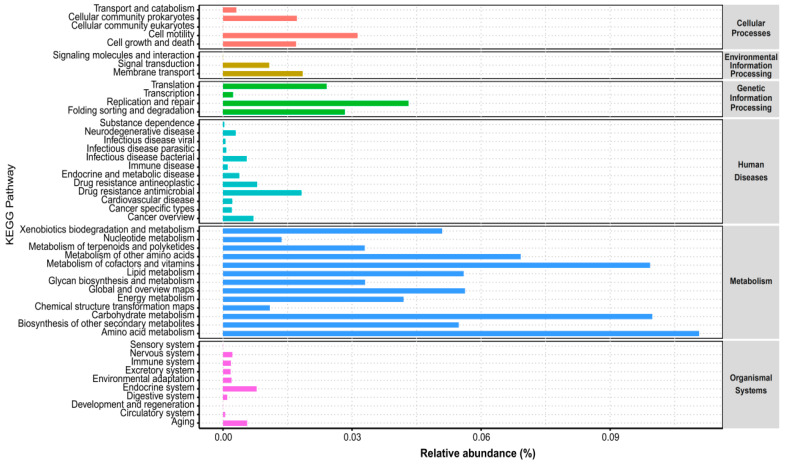
Bacterial gene functions were predicted from 16S rRNA gene-based microbial compositions using the PICRUSt2 algorithm to make inferences from KEGG annotated databases.

**Figure 6 microorganisms-10-01124-f006:**
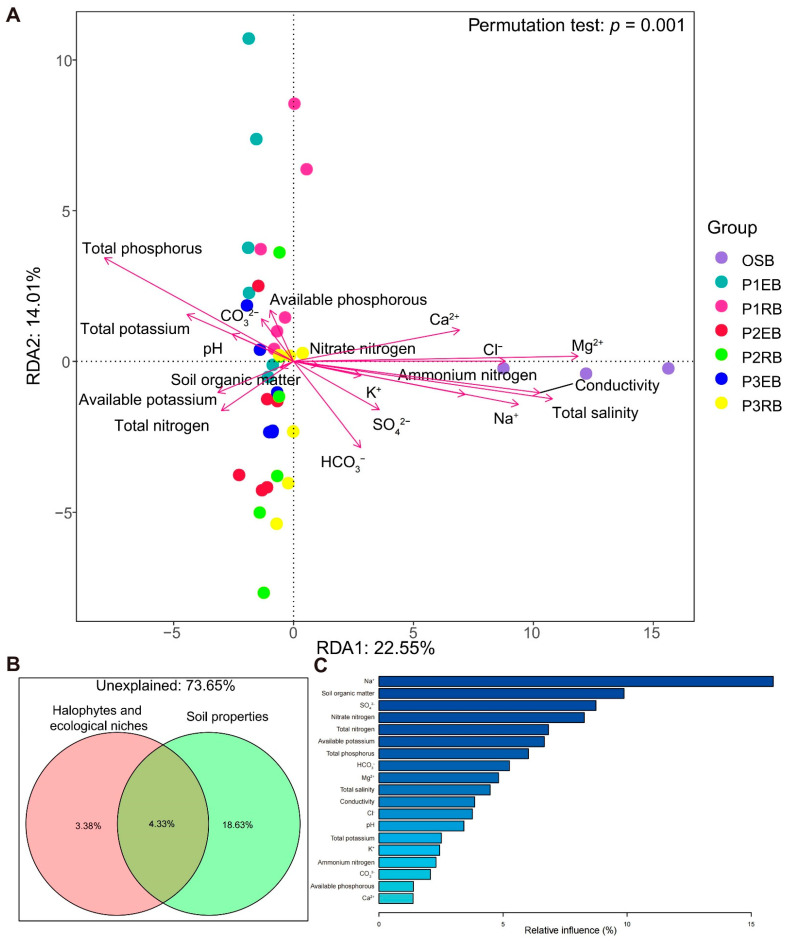
(**A**) Redundancy analysis of environmental factors and microbial community composition; (**B**) variation partitioning analysis (VPA) shows the influences of different plants and ecological niches as well as other ecological factors on the bacterial community compositions related to the studied halophytes; (**C**) the importance of environmental factors on microbial community structure was evaluated by aggregated boosted tree (ABT).

**Figure 7 microorganisms-10-01124-f007:**
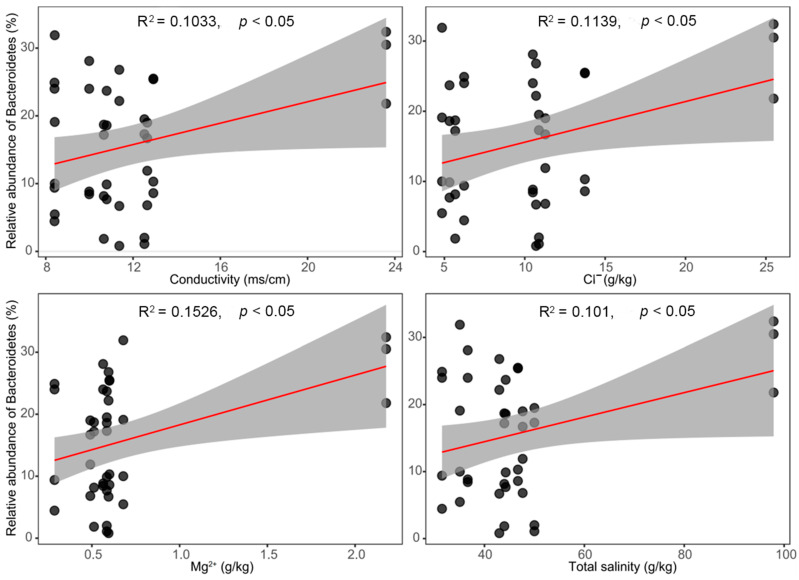
Correlation analysis between environmental factors and relative abundance of the dominant phylum *Bacteroidetes* (shading indicates the 95% confidence).

**Figure 8 microorganisms-10-01124-f008:**
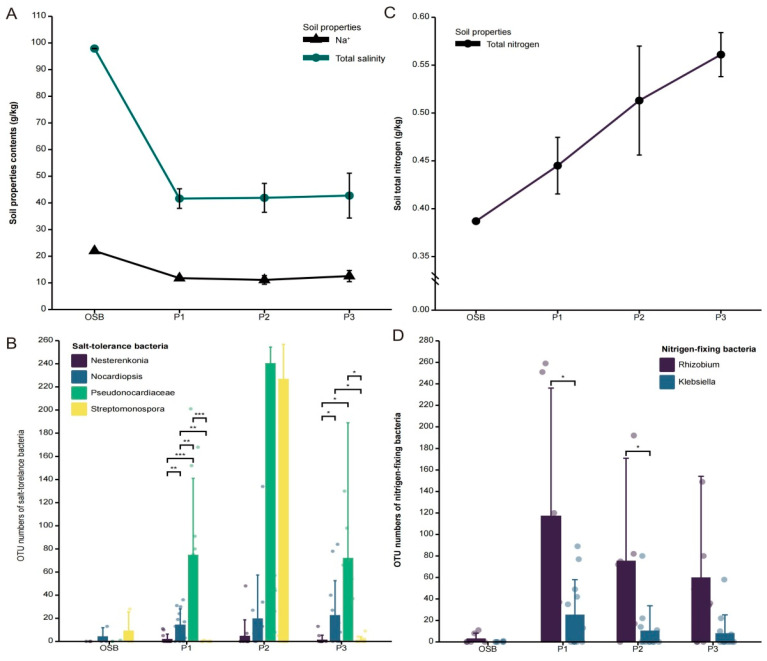
(**A**) Soil total salinity and Na^+^ contents related to three different halophytes and OSB; (**B**) the OTU abundance of the several halophilic Actinomycetes associated with three different halophytes and OSB; (**C**) total soil nitrogen content related to three different halophytes and OSB; (**D**) average abundance of nitrogen-fixing bacteria associated with three different halophytes and OSB. * *p* < 0.05; ** *p* < 0.01; *** *p* < 0.001.

**Figure 9 microorganisms-10-01124-f009:**
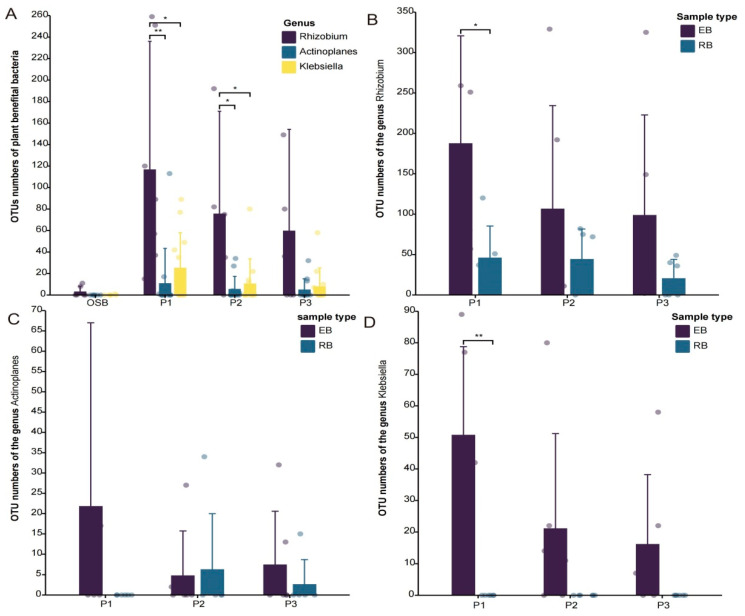
(**A**) The differences in plant beneficial bacteria related to three different halophytes and OSB; (**B**) OTU numbers of the genus *Rhizobium* associated with three different halophytes and ecological niches; (**C**) OTU numbers of the genus *Actinoplanes* associated with three different halophytes and ecological niches; (**D**) OTU numbers of the genus *Klebsiella* associated with three different halophytes and ecological niches. * *p* < 0.05; ** *p* < 0.01.

## Data Availability

The complete bacterial 16S rRNA genes V3 and V4 hypervariable region sequences generated in our study are available in the NCBI SRA database under the BioProject ID PRJNA729563 with accession number SRR14527225 to SRR14527263.

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
