# Peer review of "Bacterial Community Structure and Potential Microbial Coexistence Mechanism Associated with Three Halophytes Adapting to the Extremely Hypersaline Environment"

_microorganisms, 2022, doi:10.3390/microorganisms10061124_

Round 1
Reviewer 1 Report
The manuscript of Lei Gao and colleagues with the title “Bacterial Community Structure and Potential Microbial Coexistence Mechanism Associated with Three Halophytes Adapting to the Extremely Hypersaline Environment” analyzed the microbial diversity of endophytic and rhozospheric bacteria associated to three halophytes plants.
Comments:
Abstract:
In my opinion, it is too long.
In the abstract the authors not specify the meaning of “EB” or “RB” in the samples’ names or groups.
The authors include the number of shared and unique OTUs per sample or group but not include more information about differences of taxonomic groups between samples. Gracilimonas and Halomonas were the dominant genera in all samples?
The authors include several KEGG categories but they not discuss why these functions are relevant in the manuscript. Moreover, the functions are very general, like “metabolism”. What kind of metabolism is relevant in microorganisms associated to halophytes? For example, KEGG categories associate to human diseases are relevant in the manuscript?
Introduction:
The introduction section is so general. The authors would need to focus on the explanation of the key message of the manuscript.
Materials and Methods:
- Figure 1: low quality of the photos included.
- The authors should specify how many samples they have sequenced from each plant. In the beginning of the results section, I read that were sequenced 39 samples.
- In the section 2.1: modify the phrase “six biological samples”. I consider that the authors have 3 biological samples because they collected 3 different plants and their respective soil sample.
- Section 2.3: why the authors decided to use different set of primers for endophytic and soil samples? Could exist a bias in species composition just for the use of different set of primers?
- In the section 2.4, all sequence analysis was developed using qiime2? In qiime2, the tool implemented the ASVs clustering instead of OTUs. Why the authors did the analysis using OTUs?
- Section 2.5: how the authors built the phylogenetic tree? Using what algorithm?
- Section 2.5: the text should include more information on how the functional prediction was done with the integration of Microeco, R and STAMP.
Results:
- Section 3.1: In figure 2 I do not see that the rarefaction curves arrived to saturation but the authors included in the text that “the sequencing depths were sufficient to get the large majority of the bacterial diversity”. This phrase should be modified.
- Section 3.1: the phrase “The observed OTUs, ACE, Shannon diversity, and Simpson diversity of the bacterial communities showed significant differences between endophytic bacteria”: what kind of statistic text the authors applied to say that “showed significant differences?
- Figure 2: the authors should modify the figure in order to detect differences between OSB, EB and RB samples.
- Figure 3: In the figure text, the authors should include the meaning of NS.
- Figure 7 and 8: Genus and family names in the legend should be in Italics.
- Section 3.6: The analysis based on KEGG pathways has been mentioned in the text but the authors have not included a comparison between samples. What is the difference between endophytic and rhizosphere samples and soil?
- Figure 11: the relative abundance is based on 1 sample or all samples together?
- Section 3.7: what means nitrate nitrogen?
- Figure 14: the majority of the points are outside of the shade, then, I do not understand why the authors indicates a positive correlation between conductivity (and the others) iwht members of phylum Bacteroidetes.
Author Response
Reviewer 1: Comments
Abstract:
In my opinion, it is too long.
Reply: Thank you very much for your suggestion. We have simplified the abstract section in our newest manuscript.
In the abstract the authors not specify the meaning of “EB” or “RB” in the samples’ names or groups.
Reply: Thank you very much for your suggestion. We already modified the abstract, and we have added detailed information about the meaning of “EB” or “RB” in notes under Figure 2 in our newest manuscript.
The authors include the number of shared and unique OTUs per sample or group but not include more information about differences of taxonomic groups between samples. Gracilimonas and Halomonas were the dominant genera in all samples?
Reply: Thank you very much for your suggestion and question. We have added detailed information about the differences in taxonomic groups between samples in the section “3.2 Microbial community analysis associated with three halophytes” and Figure 2D-H in our newest manuscript. Gracilimonas and Halomonas were the dominant genera in all samples associated with three halophytes.
The authors include several KEGG categories but they not discuss why these functions are relevant in the manuscript. Moreover, the functions are very general, like “metabolism”. What kind of metabolism is relevant in microorganisms associated to halophytes? For example, KEGG categories associate to human diseases are relevant in the manuscript?
Reply: Thank you very much for your suggestion and question. We have added detailed information about the differences in nitrogen metabolism in the section “3.5 Predicted KEGG pathways of endophytic and rhizospheric bacteria based on PICRUSt2” in our newest manuscript. We found the abundance of nitrogen metabolism is universally higher in the endophytic bacterial sample groups than in rhizospheric bacterial sample groups. In our results, we can also find a relatively high abundance of membrane transport, which is relevant in microorganisms associated with halophytes to adapt to hypersaline stress.
Introduction:
The introduction section is so general. The authors would need to focus on the explanation of the key message of the manuscript.
Reply: Thank you very much for your suggestion. We have rewritten introduction in our newest manuscript.
Materials and Methods:
Figure 1: low quality of the photos included.
Reply: Thank you very much for your suggestion. We have adjusted the resolution of pictures in our newest manuscript. If necessary, we can provide the PDF version of the pictures.
The authors should specify how many samples they have sequenced from each plant. In the beginning of the results section, I read that were sequenced 39 samples.
Reply: Thank you very much for your suggestion. We have added detailed information descriptions about the number of samples in the sections “2.1 Study location and sampling methods” and “2.3 DNA extraction, PCR Amplification, 16S rRNA gene clone library construction, and Sequencing” in our newest manuscript.
In the section 2.1: modify the phrase “six biological samples”. I consider that the authors have 3 biological samples because they collected 3 different plants and their respective soil sample.
Reply: Thank you very much for your suggestion. We have revised the related description in our newest manuscript.
Section 2.3: why the authors decided to use different set of primers for endophytic and soil samples? Could exist a bias in species composition just for the use of different set of primers?
Reply: Thank you very much for your suggestion and question. The endophytic bacterial target-specific primers 335F (5'-CADACTCCTACGGGAGGC-3') and 769R (5'-ATCCTGTTTGMTMCCCVCRC-3') were designed for decreasing sequence data of mitochondria and chloroplasts.
In the section 2.4, all sequence analysis was developed using qiime2? In qiime2, the tool implemented the ASVs clustering instead of OTUs. Why the authors did the analysis using OTUs?
Reply: Thank you very much for your questions. We only used QIIME2 to cut primers as well as remove mitochondria and chloroplasts during our data processes. Other sequences processes were completed by VSEARCH and USEARCH.
Section 2.5: how the authors built the phylogenetic tree? Using what algorithm?
Reply: Thank you very much for your questions. The 16S rRNA genes ML (maximum-likelihood) phylogenetic tree was built with representative sequences of related high abundance bacteria (Top 100) using the align-to-tree-mafft-fasttree pipeline on the platform QIIME 2.
Section 2.5: the text should include more information on how the functional prediction was done with the integration of Microeco, R and STAMP.
Reply: Thank you very much for your suggestion. We have revised the related description in our newest manuscript.
Results:
Section 3.1: In figure 2 I do not see that the rarefaction curves arrived to saturation but the authors included in the text that “the sequencing depths were sufficient to get the large majority of the bacterial diversity”. This phrase should be modified.
Reply: Thank you very much for your suggestion. We also counted the Good_coverage values of all samples, which were higher than up to 0.99. Therefore, these results indicated that the sequencing depths were sufficient to get the large majority of the bacterial diversity.
Section 3.1: the phrase “The observed OTUs, ACE, Shannon diversity, and Simpson diversity of the bacterial communities showed significant differences between endophytic bacteria”: what kind of statistic text the authors applied to say that “showed significant differences?
Reply: Thank you very much for your questions. The wilcox test was used to test for significant differences among alpha diversities.
Figure 2: the authors should modify the figure in order to detect differences between OSB, EB and RB samples.
Reply: Thank you very much for your suggestion. We have added the figures and descriptions about the differences between OSB, EB and RB samples in the section “3.2 Microbial community analysis associated with three halophytes” in our newest manuscript.
Figure 3: In the figure text, the authors should include the meaning of NS.
Reply: Thank you very much for your suggestion. NS. means no significant differences among different sample groups. We have added this description under the Figure 1 in our newest manuscript.
Figure 7 and 8: Genus and family names in the legend should be in Italics.
Reply: Thank you very much for your suggestion. We have revised this mistake in our newest manuscript.
Section 3.6: The analysis based on KEGG pathways has been mentioned in the text but the authors have not included a comparison between samples. What is the difference between endophytic and rhizosphere samples and soil?
Reply: Thank you very much for your suggestion and question. We have added detailed information about the differences in nitrogen metabolism in the section “3.5 Predicted KEGG pathways of endophytic and rhizospheric bacteria based on PICRUSt2” in our newest manuscript. We found the abundance of nitrogen metabolism is universally higher in the endophytic bacterial sample groups than in rhizospheric bacterial sample groups.
Figure 11: the relative abundance is based on 1 sample or all samples together?
Reply: Thank you very much for your question. The relative abundance is based on all samples together.
Section 3.7: what means nitrate nitrogen?
Reply: The term "nitrate nitrogen" is used to refer to the nitrogen present which is combined in the nitrateion.
Figure 14: the majority of the points are outside of the shade, then, I do not understand why the authors indicated a positive correlation between conductivity (and the others) with members of phylum Bacteroidetes.
Reply: Thank you very much for your suggestion and question. Bacteroidetes could colonize a variety of habitats on earth, such as rhizospheric soil samples from various locations, including cultivated fields, greenhouse soils, unexploited areas, etc. Halophilic genus Salinibacter from the phylum Bacteroidetes always lived in salt-saturated brines and hypersaline soils. Salinibacter also was detected in our collected halophytes rhizospheric soil as well. It indicated that this taxon may represent a group of plant probiotics that can help plants cope with salt stress.

Reviewer 2 Report
The authors present a study of the bacterial community structure associated with three halophytes in Xinjiang, China. The richness and diversity of rhizospheric bacterial populations is higher than the endophytic bacteria. The manuscript is well written and the data presented seems to support the authors conclusions. I have minor comments:
1) The abstract is too long. The authors should shorten it and present only key findings.
2) page 5 in methods: The authors showed their sterilization procedure seems to eliminate viable epiphytic bacteria from the sample (however some halophilic bacteria might not grow on TSA). There is no evidence that this method can also degrade DNA present from epiphytic bacteria. Cells might not be viable but their DNA can be extracted during procedure.
3) Why archaea diversity wasn’t explored?
Author Response
The authors present a study of the bacterial community structure associated with three halophytes in Xinjiang, China. The richness and diversity of rhizospheric bacterial populations is higher than the endophytic bacteria. The manuscript is well written and the data presented seems to support the authors conclusions. I have minor comments:
1) The abstract is too long. The authors should shorten it and present only key findings.
Reply: Thank you very much for your suggestion. We have simplified the abstract section in our newest manuscript.
2) page 5 in methods: The authors showed their sterilization procedure seems to eliminate viable epiphytic bacteria from the sample (however some halophilic bacteria might not grow on TSA). There is no evidence that this method can also degrade DNA present from epiphytic bacteria. Cells might not be viable but their DNA can be extracted during procedure.
Reply: Thank you very much for your suggestion and question. We also have spread on the marine agar 2216 plates with around 2% salinity. We have rinsed 5 times with ddH2O to remove disinfectant residue and DNA that is soluble in water.
3) Why archaea diversity wasn’t explored?
Reply: Thank you very much for your question. In this study, we mainly focused on bacteria, but not archaea as well as fungi. Bacteria should be an absolute dominant group for plant microbiome and play a major role in plant microbial interaction. Therefore, the research on plant-related bacterial groups is the more important and practical guiding value.

Round 2
Reviewer 1 Report
The comments included in the first revision have been appropriately reviewed and answered. I think that the authors should sent the manuscript to an English expert editor to improve the language style.
Author Response
The comments included in the first revision have been appropriately reviewed and answered. I think that the authors should sent the manuscript to an English expert editor to improve the language style.
Reply: Thank you very much for your suggestion. We have asked an English expert to polish the language style of the article. Please check the updated revised version.
